# Innovative Carbonaceous Materials and Metal/Metal Oxide Nanoparticles for Electrochemical Biosensor Applications

**DOI:** 10.3390/nano14231890

**Published:** 2024-11-25

**Authors:** Keshavananda Prabhu Channabasavana Hundi Puttaningaiah

**Affiliations:** School of Chemical, Biological, and Battery Engineering, Gachon University, Gyeonggi-do, Seongnam-si 13120, Republic of Korea; keshavmgm@gmail.com

**Keywords:** electrochemical biosensors, carbonaceous (CBN) materials, metal nanoparticles (MNPs), metal oxide nanoparticles (MONPs), nanocomposites

## Abstract

Electrochemical biosensors have emerged as predominant devices for sensitive, rapid, and specific sensing of biomolecules, with significant applications in clinical diagnostics, environmental observation, and food processing. The improvement of inventive materials, especially carbon-based materials, and metal/metal oxide nanoparticles (M/MONPs), has changed the impact of biosensing, improving the performance and flexibility of electrochemical biosensors. Carbon-based materials, such as graphene, carbon nanotubes, and carbon nanofibers, have excellent electrical conductivity, a high surface area, large pore size, and good biocompatibility, making them ideal electrocatalysts for biosensor applications. Furthermore, M and MONPs have highly effective synergistic, electronic, and optical properties that influence signal transduction, selectivity, and sensitivity. This study completely explored continuous progressions and upgrades in carbonaceous materials (CBN materials) and M/MONPs for electrochemical biosensor applications. It analyzed the synergistic effects of hybrid nanocomposites that combine carbon materials with metal nanoparticles (MNPs) and their part in upgrading sensor performance. The paper likewise incorporated the surface alteration procedures and integration of these materials into biosensor models. The study examined difficulties, requirements, and possibilities for executing these innovative materials in practical contexts. This overview aimed to provide specialists with insights into the most recent patterns in the materials study of electrochemical biosensors and advance further progressions in this dynamic sector.

## 1. Introduction

Electrochemical biosensors have emerged as a pivotal tool in modern analytical chemistry, offering a fast, sensitive, and cost-effective approach for detecting an extensive variety of biomolecules [1,2,3]. These devices influence the standards of electrochemistry to transduce biological recognition events into quantifiable electrical signals. By coordinating biologically active components such as enzymes, antibodies, and nucleic acids with appropriate electrochemical transducers, biosensors empower the quantitative and qualitative investigation of different analytes, including glucose, proteins, and nucleic acids [2,3,4,5]. The utilization of electrochemical biosensors plays a crucial role in various sectors, including clinical diagnostics, food safety, environmental monitoring, and bioprocess control [5,6]. Despite their various benefits, the improvement of electrochemical biosensors faces several difficulties and challenges. One of the key limitations is the sensitivity and selectivity of the organic or biologically targeted components. Proteins, for example, may show restricted dependability and can be susceptible to interference from different particles [7,8]. Additionally, the immobilization of biological molecules onto the transducer surface can influence their activity and can present vague restrictions. To address these difficulties, analysts have been investigating innovative materials and fabrication advancements to upgrade the performance of electrochemical biosensors. Among the different materials investigated for overhauling biosensor performance, carbonaceous materials (CBN materials), and metal/metal oxide nanoparticles (M/MONPs) have emerged as the absolute most promising candidates due to their remarkable electrochemical and physicochemical properties [9]. This review focused on these two promising classes of materials that offer unique properties that can significantly improve the sensitivity, selectivity, and stability of biosensors.

Electrochemical biosensors depend on the specific communications and interactions between biologically targeted materials (e.g., proteins, antibodies, DNA) and target analytes to make quantifiable electrical signs. The adequacy of these biosensors generally depends upon the transducer surface, where these biological interactions occur [10,11]. Carbon-based materials, such as graphene, reduced graphene oxides (rGO), carbon nanoparticles (CNPs), carbon nanotubes (CNTs), and carbon nanofibers (CNFs), as well as metal and metal oxide nanoparticles (M and MONPs), offer specific advantages such as electrode modifiers [12,13]. Their high surface area, large pore size, excellent electrical conductivity, and compound strength and stability give an ideal platform for immobilizing bio-organic molecules and improving electron transfer between the targeted component and the electrode. Graphene, with its single-molecule-thick, two-dimensional structure, shows high electrical conductivity, mechanical strength, and biocompatibility, making it ideal for biosensor applications. CNTs have a very high surface area, large pore volume, and conductive properties that can improve the sensitivity, selectivity, and stability of biosensors. These materials offer superior electron transfer kinetics, inciting more accurate and very quick identification of biological, clinical, environmental, and food processing analytes [14,15]. In parallel, M/MONPs like gold, silver, platinum, iron oxide, titanium oxide, and zinc oxide have shown significant potential in biosensor improvement [16]. The nano-dimensional size of these materials contributes to their improved synergism and catalytic activity, considering improved signal upgrade and selectivity. MONPs can also act as carriers for biomolecules, growing the loading capacity of recognition components, while metal oxides can provide prevalent redox properties that improve the sensor’s performance.

Carbon materials known for high electrical conductivity like graphene, carbon nanotubes, and carbon black offer high stability and are productive for applications requiring rapid electron transfer. These materials also give increased surface area that enhances biomolecule adsorption, further improving sensor activity. However, carbon materials can be hydrophobic, which could confine biomolecule compatibility and require surface modifications to further develop biocompatibility. M/MONPs, including nanoparticles of metals like gold and platinum, and oxides like ZnO and Fe_3_O_4_, are highly regarded for their catalytic activity and ease of functionalization. Their tunable electronic properties allow for various applications in biosensing. However, they may show reduced stability over repeated use, as certain metal oxides can degrade, impacting long term sensor performance. Compared to other materials such as conductive polymers or enzymes, carbon materials and M/MONPs provide improved mechanical strength and electronic properties. Conductive polymers, while flexible, may lack the strength and durability expected for long term use. Enzymes, though specific and sensitive, can degrade over an extended time, which is less of a concern with carbon materials and M/MONPs. This additional setting improves the introduction by clarifying the benefits and limitations of these materials, providing readers with a balanced view of their sensibility for various biosensor applications. Carbon materials and metal/metal oxide nanoparticles (M/MONPs) are distinct in their properties, designs, and mechanisms of interaction with analytes in biosensing applications. Carbon materials, including graphene, carbon nanotubes (CNTs), carbon nanoparticles and carbon dark, have broadened sp^2^ hybridized networks and show high surface regions because of their layered or cylindrical designs. This structural composition empowers high conductivity and gives a steady structure to biomolecule immobilization. And these materials are intrinsically conductive, particularly graphene and CNTs, due to their delocalized π-electrons. This property works with quick electron move during redox responses and gives a high signal- to-noise proportion in biosensors. M/MONPs, like gold (Au), silver (Ag), platinum (Pt), and metal oxides like iron oxide (Fe_3_O_4_), have crystalline or polycrystalline designs with different oxidation states. These nanoparticles show specific surface properties and electrochemical redox abilities because of the presence of metallic components, empowering catalytic activities that are significant for improving biosensor sensitivity. These materials are chemically reactive and can catalyze explicit biochemical responses, considering specific biomolecule identification. However, their reactivity can lead to prompt stability issues, such as agglomeration or oxidation in biological environments, which may degrade the sensor.

This review aimed to provide a thorough outline and comprehensive overview of recent advances in the improvement of carbon-based and MONPs-hybrid electrochemical biosensors. We will explore the modification techniques, functionalization methodologies, and sensor fabrication strategies that have been utilized to improve and develop sensor performance. In addition, we will analyze the challenges and future perspectives in this rapidly developing field, highlighting the ability of these materials to revolutionize biosensing advancements across different industries. In summary, this review will provide significant experiences in the innovative use of CBN materials and M/MONPs in electrochemical biosensors, featuring their critical role in the next generation of biosensor stages for clinical healthcare monitoring, environmental observing, food processing and beyond.

## 2. Advantages and Applications of Carbonaceous Materials, Metal, and Metal Oxide NPs

CBN materials, including graphene, rGO, CNTs, carbon black, and activated carbon, have great attention due to their high electrical conductivity, versatility, huge surface area, large pore volume, mechanical strength, and superior electrochemical properties [17,18,19,20]. These materials can be used as both the transducer and the assistance for biologically targeted components, giving a good environment for electron transfer and upgrading the sensitivity of biosensors. CNTs are cylindrical-shaped carbon structures that exhibit high electrical conductivity and mechanical properties [18]. They can be used as conductive wires to defeat any barrier between the targeted component and electrode surface and improve electron transfer kinetics. Also, CNTs can be functionalized with various groups to improve their biocompatibility and the immobilization of biological particles. Graphene is a single layer of carbon atoms coordinated in a hexagonal matrix. It has exceptional electrical conductivity, a high surface area, and radiant mechanical strength [19]. Graphene can be used as a conductive substrate for the immobilization of bio-organically designated parts, giving a consistent and conductive platform for electron transfer. In addition, graphene can be modified with different functional groups to upgrade its biocompatibility and selectivity. Carbon black is a fine, amorphous carbon material widely used in many applications due to its larger surface area and excellent electrical conductivity. Its properties make it ideal as a conductive support for detecting bioactive components in biosensors. Additionally, carbon black can be modified with different functional groups to improve its biocompatibility and enhance the immobilization of biological particles [20,21].

M and MONPs have emerged as promising materials for electrochemical biosensors because of their extraordinary properties, including high surface-to-volume proportion, tunable electronic properties, and catalytic activity [17,20]. These nanoparticles (NPs) can be used as electrocatalysts to enhance the sensitivity and selectivity of biosensors. Additionally, they can be functionalized with biologically targeted components to make highly sensitive and specific biosensors. MONPs (e.g., TiO_2_, ZnO, and Fe_2_O_3_) offer numerous properties, including high surface area, synergist action, and photocatalytic properties [22,23,24]. They can be used as electrocatalysts to enhance the sensitivity and selectivity of biosensors, especially for the recognition of redox-active analytes. Moreover, MONPs can be functionalized with bioactive components to make profoundly sensitive and highly selective biosensors.

To further improve the performance of electrochemical biosensors, specialists have researched the combination or hybridization of CBN materials and M/MONPs. Hybridization of materials can offer synergistic effects, combining the advantages of the two classes of materials. For example, the hybridization of CNT with MONPs can upgrade the electron transfer kinetics and improve the sensitivity and selectivity of biosensors with a lower limit of detection range. Similarly, the combination of graphene with MONPs can give a consistent, steady, and conductive platform for the immobilization of biologically active components and improve the catalytic activity of the biosensor [25,26,27,28].

## 3. Carbonaceous Materials, Metal, and Metal Oxide NPs for Sensor Applications

### 3.1. Surface Modification Techniques

The performance of electrochemical biosensors generally relies upon the adjustment of their electrode surfaces, which enhances the electron transfer kinetics and sensitivity of target analytes. Surface modification is a basic and crucial step toward biosensor design, as it ensures the steady and specific collaboration between the electrode and the biomolecules of interest. The utilization of CBN materials and M/MONPs has significantly advanced the development of electrochemical biosensors. Figure 1 summarizes the different surface modification procedures that can be applied to these materials, focusing on their application in improving sensor performance.

Drop-casting is one of the simplest and most widely used strategies for the modification of electrode surfaces with NPs. In this strategy, a suspension of carbon-based materials, for example, graphene, CNTs, carbon dark, and MNPs is dropped onto the electrode surface, followed by drying under ambient conditions. The flexibility of this strategy lies in its ability to deposit a uniform layer of the detecting material without requiring sophisticated equipment. Graphene-based NPs, for instance, have been frequently utilized through drop-casting to improve electrochemical sensitivity because of their excellent electrical conductivity and enormous surface area [27,28,29,30].

Moreover, the combination of CNP with MNPs (like gold, platinum, or metal oxides like ZnO and TiO_2_) improves both sensitivity and selectivity in biosensing applications. These techniques are very simple, and accessibility requires no complex equipment, making it easy to implement and allowing the deposition of a uniform sensing material layer. It can incorporate various materials, including graphene, carbon nanotubes (CNTs), carbon black, and metal nanoparticles (MNPs). It is very effective at improving electrochemical sensitivity due to the high conductivity and large surface area of carbon-based materials. The disadvantages are its inconsistent reproducibility and lack of precise control over layer thickness and uniformity. Occasionally, during the drying process, nanoparticles may aggregate, thereby influencing the performance of the sensor.

Electrodeposition is another popular well-known strategy for surface modification, including the electrochemical reduction of metal salts to frame a nanostructured layer on the electrode surface. This strategy offers precise control over the thickness and morphology of the deposited material, making it ideal for integrating M/MONPs with carbon-based electrodes. Electrodepositions can be customized to produce composite films that offer high catalytic action and biocompatibility, both of which are crucial and significant for biosensor applications. For instance, electrodeposited gold nanoparticles (AuNPs) on graphene-altered electrodes have shown excellent outcomes in detecting biomolecules, such as glucose and dopamine [31,32].

Also, metal oxides like ZnO and Fe_3_O_4_ are electrodeposited to upgrade electron transfer kinetics, frequently offer higher sensitivity, and lower recognition limits for various analytes. The main advantage of this technique is that it allows for the controlled deposition of nanostructured films with tunable thickness and morphology and increases electron transfer kinetics, which is crucial for biosensor sensitivity. It is ideal for applications requiring integration of M/MONPs with biomolecules on carbon-based electrodes. The disadvantage of this is that it requires an electrochemical workstation, limiting accessibility, and precise electrodeposition may require longer deposition times. Sometimes the over-deposition can lead to decreased sensitivity due to thicker films.

Layer-by-layer (LbL) assembly is a flexible surface modification procedure that takes into consideration the successive deposition of oppositely charged materials, framing complex designs on the electrode surface. This strategy is especially valuable for developing biosensors with customized structures, as it empowers the precise arrangement of nanomaterials and biomolecules. LbL assembling has been utilized to immobilize MNPs and carbon-based materials like graphene oxide and CNTs onto the electrode surface. The alternating deposition of these materials results in enhanced electron transfer and a high surface-to-volume proportion, improving the sensor’s capacity to identify target analytes with high sensitivity [32,33,34].

Moreover, this method allows for the incorporation of biorecognition components, like enzymes or antibodies, inside multilayered films, giving a stable environment for their movement activity and improving biosensor performance. The advantages of this method are customizable structure. This allows sequential deposition of oppositely charged materials, achieving complex multilayered structures and provides a high surface-to-volume ratio, ideal for high-sensitivity biosensing. It enables the stable incorporation of enzymes, antibodies, and other biomolecules. The disadvantage of this method is that it is a complex process. It requires multiple deposition steps, which can be time intensive. Multilayer structures may experience reduced stability, depending on environmental conditions.

Self-assembled monolayers (SAMs) are highly ordered molecular assemblies formed on the electrode surface through the spontaneous adsorption of functionalized particles, typically thiol-based compounds on gold electrode. SAMs provide a platform for additional functionalization with nanomaterials and biomolecules. In biosensor applications, SAMs can upgrade the immobilization of biomolecules while preventing nonspecific adsorption, subsequently improving the selectivity and stability of the sensor. By coordinating M/MONPs into SAM-modified electrodes, researchers have accomplished huge upgrades in the sensing and detection of different biomolecules. For example, AuNPs implanted in SAMs have been utilized to make profoundly sensitive glucose sensors [35]. The advantages of this method are high selectivity and stability. The SAMs prevent nonspecific adsorption and enhance selectivity. It can be customized with specific functional groups to immobilize biomolecules, allowing for precise modification of electrode surface properties. Integration of AuNPs or other MNPs enhances sensor response, particularly for glucose and other small-molecule biosensors. The main disadvantage is that the SAMs may degrade over time, especially in harsh conditions. Stability and functionality may vary with pH, temperature, and ionic strength. It forms monolayers, limiting the amount of functional material that can be immobilized.

Covalent functionalization includes the arrangement of strong chemical bonds between the electrode surface and adjusting specialists. This strategy offers robust attachment of NPs to the electrode, guaranteeing long-term stability and high sensitivity in biosensing applications. Carbon-based materials like graphene and CNTs are frequently covalently functionalized with biomolecules or MNPs to improve their electrochemical properties. For instance, a covalent connection of enzymes or antibodies to a CNT-modified electrode has been demonstrated to significantly upgrade biosensor performance for the detection of proteins, glucose, and DNA [36,37,38].

Moreover, covalent functionalization considers precise control of surface chemistry, guaranteeing reproducibility and long-term stability in biosensor applications. The advantages of this method are that strong covalent bond formation offers high stability, making sensors more durable and reusable and functionalization with biomolecules (enzymes, antibodies) improves selectivity for specific analytes. It provides a stable and repeatable surface modification and boosts electron transfer rates, ideal for high-performance biosensors. The main disadvantage of this method is the complex functionalization process. It requires specific reagents and conditions, which can increase time and cost and the covalent binding may alter the activity of immobilized biomolecules.

Plasma treatment is an effective surface modification technique that presents functional groups onto the electrode surface, improving its reactivity and compatibility with biomolecules. Carbon-based electrodes, such as graphene and CNTs, can be modified with oxygen-containing groups through plasma treatment, improving their dispersibility and the immobilization of biomolecules. This method has been utilized to adjust graphene oxide and CNTs with functional groups that advance the connection of MNPS or biomolecules. For example, plasma-treated CNTs have been utilized to immobilize compounds for glucose recognition, prompting further developed sensitivity and quicker response times [39]. The advantage of this method is strong enhanced surface reactivity. This adds functional groups to carbon-based electrodes, improving compatibility with biomolecules; plasma-treated surfaces often show enhanced immobilization of biomolecules, increasing sensitivity. This allows for the modification of different materials (e.g., graphene oxide, CNTs) with various functional groups, improving biosensor design. This quick treatment time makes it efficient for large-scale applications. The main disadvantage of this method is the specialized equipment needed for the plasma treatment. It requires access to plasma generators, which can be costly, and sometimes this high-energy plasma may cause damage to sensitive materials, affecting performance, and precise control over functional group density and uniformity can be challenging.

Overall, surface modification strategies play a crucial role in improving the performance of electrochemical biosensors. The integration of CBN materials and M/MONPs via methods such as drop-casting, electrodeposition, LbL assembly, SAMs, covalent functionalization, and plasma treatment, has prompted critical advancements in biosensor sensing, selectivity, and stability. These techniques provide a platform for the improvement of next-generation biosensors with further developed performance for a wide range of applications in clinical diagnostics, environmental monitoring, and food safety.

### 3.2. Carbonaceous Materials (CBN Materials)

CBN materials are reliable materials for biosensor applications owing to their affordable electrical properties, robust mechanical strength, large surface area, bioreceptor immobilization, and biocompatibility [40]. CBN materials are employed in various biosensor devices that manifest binding sites for target biomarkers to facilitate their capture and identification. Moreover, CBN materials convert the molecular interactions detected on the electrode surface into measurable electrical signals, enabling quantitative analysis. Furthermore, they increase signal amplification and sensitivity [41]. However, efficient signal capture for biomolecular recognition remains a challenge in the development of CBN materials. The CBN materials most utilized in biosensors include CNTs, graphene, nanodiamonds, and fullerenes. Some CBN materials are directly integrated into the bare electrodes, serving a dual role in analyte identification and signal transduction [42]. Recently, Kim et al. [43] developed a single-walled CNT (SWCNT)-based immunosensor for CD^4+^ T cell detection by addressing the problems of immobilization efficiency, reproducibility, and linear quantification. Through repeated two-step O_2_ plasma treatment with recovery periods, the team achieved enhanced CNT surface functionalization and enabled linear electrochemical signal generation proportional to the number of bound cells. The SWCNT electrode exhibited a slope of 4.55 × 10^−2^ μA/dec within the target range of 10^2^~10^6^ cells/mL, with a detection limit of 1 × 10^2^ cells/mL. Reduced CD^4+^ T cell counts are markers of HIV progression; this immunosensor offers simple, low-cost point-of-care diagnostics, potentially aiding in HIV patient management. Wang et al. [44] studied a novel electrochemical sensor for hypering detection that was developed using a 3DG-MWCNTs network fabricated directly on a glassy carbon electrode (GCE) using the pulse potential method. In contrast to bare GCE and 3DG-modified electrodes, the 3DG-MWCNTs complex exhibited an increased response toward hyperin redox, which was attributed to the large surface area and excellent conductivity of the 3DG-MWCNTs, facilitating efficient electron transfer. The sensor portrayed a wide linear range of 5.0 × 10^−9^ to 1.5 × 10^−6^ mol L^−1^ and a low detection limit (LOD) of 1.0 × 10^−9^ mol L^−1^ with high sensitivity for hyperin detection. Additionally, the sensor displayed remarkable stability, selectivity, and reproducibility in detecting hyperin in various matrices. Similarly, Chaitra et al. [45] designed biomass-based carbon nanospheres using “touch-me-not” (*Mimosa pudica*), which was coated on carbon fiber paper and employed as a host matrix for palladium NPs via electrochemical deposition. The modified electrodes exhibited excellent electrocatalytic activity toward morin oxidation, with a sensitive peak at −0.30 V (vs. saturated calomel electrode) and a LOD of 572 fM, within a linear response range of 37.50–130 pM. The sensor successfully analyzed the morin content in mulberry and guava leaves, demonstrating its potential for the sustainable approach of using biomass for sensor fabrication. Similarly, various studies have been conducted on the fabrication of biomaterial/CNT hybrid systems. Through a potent acid treatment, the ensuing development of carboxylic and phenolic groups on protein-associated CNTs along with nucleic acid-functionalized CNTs was observed. This chemical shortening of the SWCNTs’ nanotube electrodes facilitated the covalent immobilization of biomaterials, proteins, and nucleic acids onto the surface of the SWCNTs [46]. An electrochemical biosensor was devised to detect the human carcinogen aflatoxin B1 by employing MWCNTs deposited on indium/tin/oxide (ITO) electrodes [47]. Figure 2 illustrates the various methods used for fabricating carbon-based materials, which are integral to biosensor applications. The versatility, conductivity, and biocompatibility of carbon-based materials render them highly suitable for biosensor development [48,49].

Figure 3 shows a detailed step-by-step plan for making an rGO-coated electrochemical immunosensor specifically designed to detect cancer antigens. The fabrication process involves several key stages, each of which contribute to the creation of a highly selective and specific biosensor. Although CBN materials-based biosensors are promising, they face practical challenges in biological applications. Notably, biosensor fabrication requires a specific size and helicity of CBN materials, posing difficulties in size control during synthesis [50]. Moreover, the cost-efficiency challenges of large-scale production hamper the widespread use of high-purity nanomaterials, contributing to the prohibitively expensive CBN materials and limiting their practical commercial applications. In carbon-based biosensors, the immobilization of enzymes on CBN materials-coated electrodes raises concerns about potential damage to biological receptors, affecting biocompatibility, biological activity, and structural stability [51]. Addressing this issue requires the development of reversible, reusable, and enduring systems, as opposed to the prevalent use of irreversible, disposable, and single-use devices, which present an ongoing challenge in this field. Furthermore, the functionalization and conjugation of CBN materials with organic compounds or metallic NPs can impart novel properties, including physical, chemical, mechanical, electrical, and optical attributes [52]. This augmentation further enhances their suitability for biosensor applications. Recently, Keshavandaprabhu et al. [53] developed an organic compound-based polymeric cobalt Pc (poly (CoTBImPc)) complex with tetrabenzimidazole (TBIm) substitutions for hydrogen peroxide (H_2_O_2_) sensing. Poly (CoTBImPc) showcased distinctive redox properties with peaks corresponding to Co^2+^/Co^1+^ and Pc^II−^/Pc^III−^ redox couples. To enhance its conductivity, surface area, and sensitivity, the complex was coated onto a CNT-modified GCE (GCE/CNT/poly (CoTBImPc)). This brought about a highly delicate amperometric sensor for H_2_O_2_ that showed a linear response in the 10–100 nM range and a remarkable LOD of 2 nM. Also, the sensor showed outstanding reproducibility, repeatability, and long-term stability without any deficiency of catalytic activity. Remarkably, it maintained high selectivity for H_2_O_2_ even within the presence of expected potential interfering species, exhibiting its dependability for practical applications. Imadadulla et al. [54] integrated and characterized novel CoTBIPc with TBIm. Electrochemical examinations revealed the redox conduct of the Co(II)/Co(I) metal center. CoTBIPc went through electropolymerization on the GCE, shaping film (GCE/poly (CoTBIPc)) for H_2_O_2_ detection. To improve the surface area and sensitivity, CoTBIPc was electropolymerized on GO-covered GCE/GO/poly (CoTBIPc). This hybrid material showed improved catalytic activity with a linear H_2_O_2_ sensing range of 2 to 300 μM contrasted with 3 to 160 μM of GCE/poly (CoTBIPc). Amperometric sensors utilizing both the modified electrodes showed linear response with LODs of 0.8 μM and 0.6 μM for H_2_O_2_ sensing, respectively. GCE/GO/poly (CoTBIPc) offered predominant reproducibility, repeatability, and stability while maintaining high selectivity for H_2_O_2_. This study showed the potential of CoTBIPc-based electrodes for H_2_O_2_ detection with improved performance. Recent studies utilized materials based on their mechanical and electrical properties for biosensor applications via the amalgamation of CNPs. The integration of CNPs with biotechnology provides an economical method for the real identification of targets and leverages specific antibody recognition in biosensor development. Figure 4 shows the design and fabrication steps for two types of electrodes, CNT-modified and GO-modified electrodes, for the electrochemical detection of H_2_O_2_. Both methods exploit the unique properties of CBN materials to enhance the sensitivity and performance of the sensors in H_2_O_2_ detection [53,54]. Table 1 provides an overview of different CBN materials, their unique properties, and their suitability for detecting various analytes in biosensor applications [55,56,57,58,59,60,61,62,63,64,65,66].

### 3.3. Metal and Metal Oxide NPs

MNPs act as “electron wires” to transport the electrons produced in the bioreaction to sensing electrodes [66]. The integration of MNPs into biosensors enhances the functionality of biosensor devices. MNPs serve as transducers because of their unique physiochemical properties in the 1 to 100 nanoscale range, providing a substantial surface-to-volume ratio and facilitating the sensing and detection of biomolecules [67]. Various MNPs such as Pb, Pd, Cu, and Se NPs, and metal oxide NPs, such as zinc oxide (ZnO), iron oxide (Fe_2_O_3_), copper oxide (CuO), titanium dioxide (TiO_2_), indium oxide (InO_2_) NPs, have been used in sensing applications [68,69]. Functionalization of NPs with specific ligands or biomolecules on the surface imparts selectivity for the target analyte, harnessing the unique properties of MNPs such as their high surface area, catalytic activity, and optical properties, making them suitable candidates for sensing [70]. Among the various MNPs, gold, silver, and platinum have been extensively studied because of their well-established chemical inertness on a macroscopic scale. Despite their chemical inertness in larger dimensions, these noble metals exhibit unique physicochemical characteristics when scrutinized at the nanoscale. Guo et al. [71] studied an amperometric glucose biosensor using both Rh and Au NPs. The Rh NP-modified Pt electrode contributed substantially to the surface area, with numerous active sites enhancing the electrocatalytic activity. Furthermore, Au NPs strategically positioned near the active regions of the enzyme facilitated the oxidation of peroxide molecules, thereby amplifying the sensitivity of the glucose sensor. The functionalization of Rh and Au NPs played a crucial role in optimizing the biosensor performance for efficient glucose detection. Additionally, the functionalization of MNPs is defined by the physical or electrochemical changes that occur after biomolecular analytes bind to the immobilized receptor target on the surface of the MNP. MNPs exhibit various functions, including immobilizing platforms, accelerating electron transfer, catalyzing chemiluminescent reactions, amplifying changes in mass, and enhancing refractive index modification [72]. Figure 5 gives an example of the fabrication and modification of ZnO NPs and their resulting amperometric reactions when deposited on carbon paper. The correlation of various ZnO samples, such as potassium chloride (KCl), pristine, ammonium fluoride (NHF)-doped, and ethylenediamine (EDA)-hybrid ZnO, exhibits the effect of different modifications on the electrochemical performance, with the EDA modification showing the most pronounced advantages [73].

Zhu et al. [74] investigated the fluorescence quenching properties of silver-coated gold triangular nanoplates (Ag/Au TNPs) for detection of carcinoembryonic antigen (CEA). SPR-incited nonradiative energy transfer from CEA to the Ag/Au TNPs essentially extinguished CEA fluorescence, showing a higher extinguishing efficiency than that of bare gold TNPs. This impact escalated with expanding silver covering thickness and CEA focus, with a LOD of 5 pg/mL for the CEA, which is a demonstration of the upgraded sensitivity of the Ag/Au TNPs. Furthermore, Lan et al. [75] developed a chemiluminescence (CL) course through biosensors for glucose recognition by immobilizing glucose oxidase (GOD) and horseradish peroxidase (HRP) on eggshell films utilizing glutaraldehyde. The CL process involved enzymatic glucose oxidation, producing D-gluconic acid and H_2_O_2_, with H_2_O_2_ oxidizing luminol to emit CL in the presence of HRP. The immobilization conditions (time, GOD/HRP ratio, and glutaraldehyde concentration) were thoroughly studied. The biosensor exhibited stable storage at 4 °C for 5 months, with quick response, high sensitivity, simple operation, and assembly. Successfully applied for human serum glucose determination, this biosensor holds promise for practical applications.

Lei et al. [76] designed an electrochemiluminescence (ECL) cytosensor for highly sensitive and selective cytosensing of K562 cancer cells. The research group immobilized PtNPs on CNTs on the WE, which enhanced both the electronic transmission rate and surface area, as shown in Figure 6. The electrode was further modified with aptamers for the precise capture of cancer cells. The researchers also devised a novel class of nanoprobe by integrating concanavalin A (Con A) for specific recognition and signal amplification using Au cage/Ru (bpy)32+ nanostructures. Operating on a sandwich-type cytosensor format, the ECL signals increased proportionally with the quantity of K562 cancer cells, reflecting the augmented presence of Au cage/Ru (bpy)32+ labelled Con A. The cytosensor demonstrated outstanding analytical performance across a broad detection range of 500 to 5.0 × 10^6^ cells mL^−1^, and a detection limit of 500 cells mL^−1^. Additionally, the method exhibited high precision, longer stability, and reproducibility. Similarly, Liang et al. [77] presented a highly sensitive self-enhanced ECL cytosensor for monitoring cell apoptosis, employing ruthenium/silica composite nanoparticles (Ru-N-SiNPs) labeled with annexin V as signal probes. The as-designed Ru-N-SiNPs resulted in enhanced ECL intensity and higher emission efficiency, which were attributed to a shorter electron transfer path and reduced energy loss. The developed ECL cytosensor successfully evaluated the efficacy of paclitaxel against MDA-MB-231 breast cancer cells within a concentration range of 1 to 200 nM. The detection limit was determined to be 0.3 nM, and the correlation coefficient was 0.9917, indicating improved accuracy.

Optimization such as adjusting synthesis methods, conditions, or materials relies mainly on the synthesis of various metal/metal oxide NPs. Gold, silver, platinum, copper, palladium, lead, selenium, and metal oxides have been synthesized using chemical, physical, and biological synthesis methods [78]. Chemical methods provide control over size and shape. Physical methods such as mechanical milling and vapor deposition furnish a controlled surface morphology and crystal structure. Additionally, biological methods such as green synthesis employ plants, algae, fungi, and microorganisms for NP synthesis, to produce biocompatible NPs with biomolecule encapsulation further enhancing biocompatibility and reducing toxicity. Biological methods are cost-effective and scalable. Furthermore, the cellular uptake of MNPs is influenced by the physicochemical properties, and the biocompatibility involves both qualitative and quantitative analyses [79]. Despite these synthesis methods, it still faces many challenges, such as precisely controlling particle size and morphology, ensuring high purity and reproducibility, and achieving large-scale production without compromising environmental friendliness and cost-effectiveness. Moreover, surveying the effect of physicochemical properties on cellular uptake and creating hearty biocompatibility evaluations are significant for tending to potential toxicity concerns. Recently, a synergistic combination of MNPs with polymers or CBN materials has shown promise in biosensor applications. This study was conducted by Liu et al. [80], who investigated the synergistic impacts by using two distinct kinds of noble MNPs in an electrochemical biosensor, showing upgraded electron movement. This study revealed that Pd/Co amalgam NPs implanted in CNFs display predominant analytical abilities, especially for hydrogen peroxide and nitrite detection. This synergistic impact appeared as a diminished overpotential and significantly higher decrease current for H_2_O_2_, as well as an expanded oxidation peak and diminished overpotential in the nitrite cyclic voltammograms. This highlights the viability of utilizing a combination of noble MNPs to improve the performance of electrochemical biosensors for recognizing explicit analytes. Table 2 gives a detailed overview of the capacities of different MNPs for the sensing of various analytes. Every section in the table highlights the MNP utilized, the objective analyte, the detection technique utilized, the sensitivity and detection limits achieved, and the remarkable highlights that contribute to their effectiveness. This demonstrates the significant role of MNPs in enhancing the performance of biosensors in various analytical applications [81,82,83,84,85,86,87,88,89,90]. Table 3 outlines the key properties of various CBN materials and M/MONPs in electrochemical biosensor applications. This table highlights each material’s selectivity, detection limits, sensitivity, and technical benefits to help readers compare based on application needs.

## 4. Conclusions, Challenges, and Future Perspectives

The improvement of CBN materials and M/MONPs has fundamentally advanced the field of electrochemical biosensors. These materials offer outstanding properties that can improve the sensitivity, selectivity, and stability of biosensors, engaging their application in a wide range of fields. By consolidating these materials with advanced modification procedures and biologically active components, researchers are persistently pushing the boundaries of electrochemical biosensor advancement, planning for new and intriguing applications.

In this article, we have explored the promising capability of creative CBN materials and M/MONPs for electrochemical biosensor applications. These materials offer remarkable properties that can essentially improve the sensitivity, selectivity, and dependability of biosensors, empowering their application in many fields. CBN materials, like CNT, graphene, and carbon dark, give high electrical conductivity, expanded surface area, and mechanical strength, making them ideal for use as the two transducers and supports for biologically targeted materials. M/MONPs offer high catalytic action, surface-to-volume proportion, and tunable electronic properties, making them sensible for use as electrocatalysts and for the immobilization of target natural dynamic materials. By joining these materials through hybrid structures, analysts can use their synergistic effects to make essentially additionally developed biosensors. These materials can offer high electron transfer kinetic energy, further develop biocompatibility, and upgrade synergist action, inciting improved biosensor performance.

Despite significant progress in electrochemical biosensor development, a few difficulties still need to be addressed. Vague adsorption of biomolecules onto the biosensor surface can obstruct the identification of the goal analyte. This can incite lessened or diminished sensitivity and selectivity. Keeping up with the dependability of biosensors after some time is vital for their functional application. Factors such as temperature, pH, and capacity conditions can impact the solidity of both the organically designated materials and the transducer materials. The fabrication of electrochemical biosensors can be complex and costly, confining their wide adoption. Creating cost-effective production methodologies is crucial for making biosensors more available. For point-of-care applications, biosensors should be little and advantageous. Developing miniaturization methodologies while keeping up with responsiveness and selectivity is a challenge.

To address these difficulties and further develop the field of electrochemical biosensors, future research should focus on refining synthesis methods for improved homogeneity and functionalization strategies that enhance biocompatibility. Exploring the use of novel materials, such as metal/organic frameworks (MOFs) or transition metal dichalcogenides (TMDs), in conjunction with carbon materials could open new avenues for enhancing electrochemical reactions and biosensing capabilities. Further interdisciplinary collaborations between materials science, biology, and engineering will be crucial in developing next-generation biosensors that are not only more sensitive and selective but also capable of real-time monitoring in complex biological environments. The progression of new materials with modified properties and imaginative assembling techniques can improve the performance and cost-adequacy of biosensors. Procedures for forestalling or directing biofouling, for example, surface changes or the use of antifouling coatings, are crucial for improving the long-term stability and dependability of biosensors. Research on procedures for improving the stability of biologically designated materials and transducer materials can extend the lifetime of biosensors. Making adaptable and practical fabrication strategies, such as, printed or 3D-printed biosensors, can make biosensors more available. Joining electrochemical biosensors with various advances, for example, microfluidics, remote correspondence, and artificial intelligence, can make more modern, adaptable, flexible, and sophisticated diagnostic devices.

In conclusion, electrochemical biosensors hold extraordinary promise for various applications. By tending to the current challenges and investigating new materials and manufacturing strategies, researchers can continue to advance the field and create biosensors that are more sensitive, specific, stable, cost-effective, and reasonable. The future of electrochemical biosensors is bright, and their impact on society is likely to be significant.

## Figures and Tables

**Figure 1 nanomaterials-14-01890-f001:**
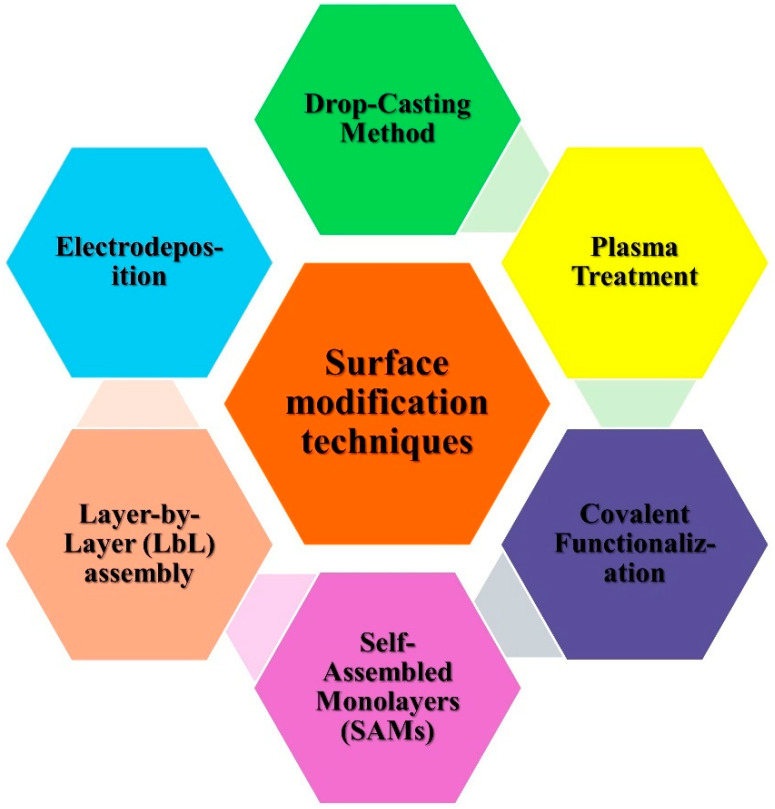
Different types of surface modification techniques.

**Figure 2 nanomaterials-14-01890-f002:**
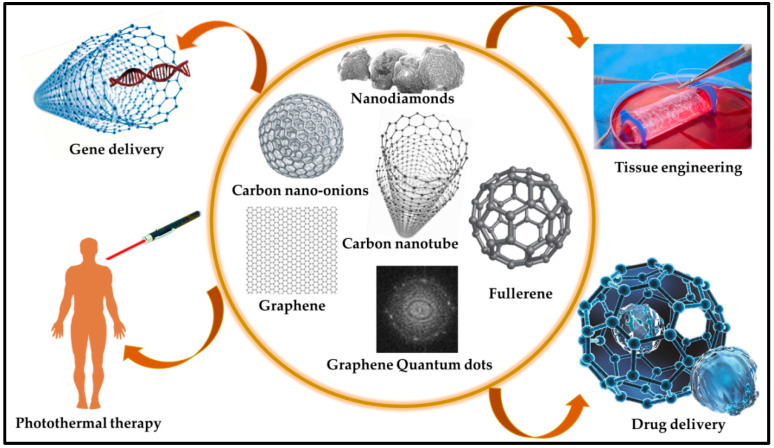
Various applications of carbon-based materials in different fields [48].

**Figure 3 nanomaterials-14-01890-f003:**
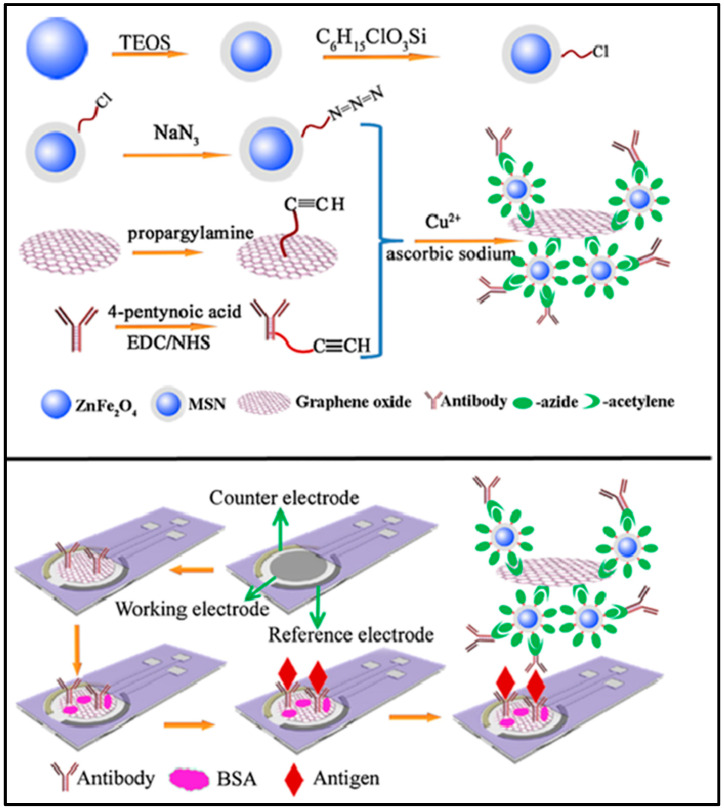
Schematics of the fabrication of magnetic silica/graphene oxide (MSN/GO)-coated electrochemical immunosensor for the detection of cancer antigen 153 (CA 153). Reproduced with permission [49] Copyright 2014 from Elsevier Publications.

**Figure 4 nanomaterials-14-01890-f004:**
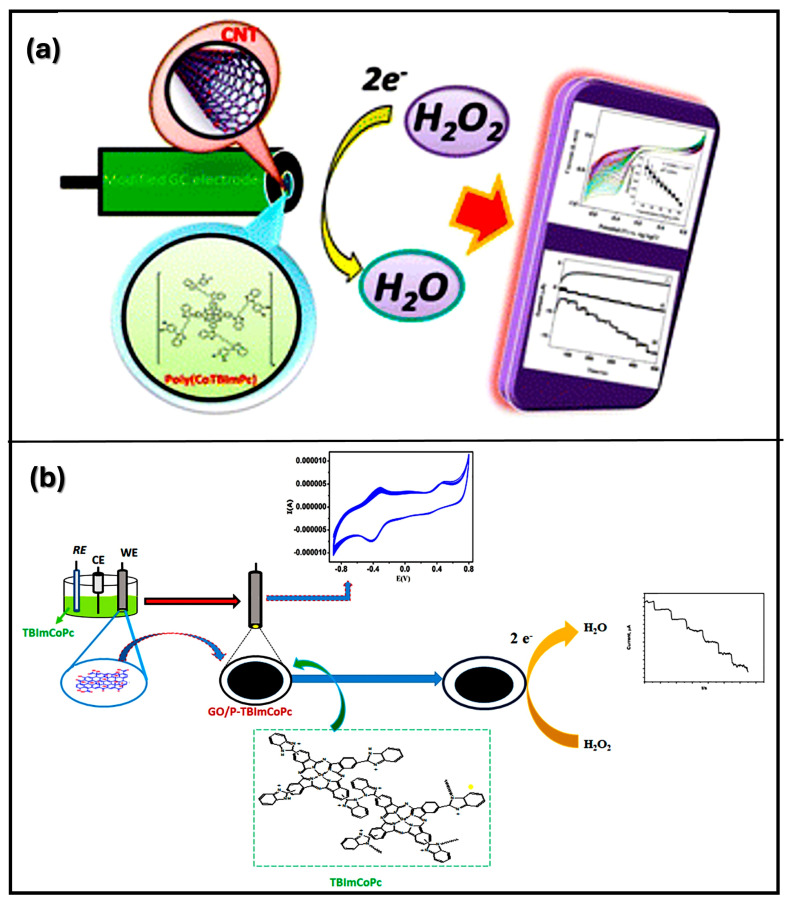
The fabrication of CNT and GO for detection of H_2_O_2_ (**a**) CNT-modified electrode [53], and (**b**) GO-modified electrode for detection of H_2_O_2_. Reproduced with permission [54] Copyright 2018 from Elsevier Publications.

**Figure 5 nanomaterials-14-01890-f005:**
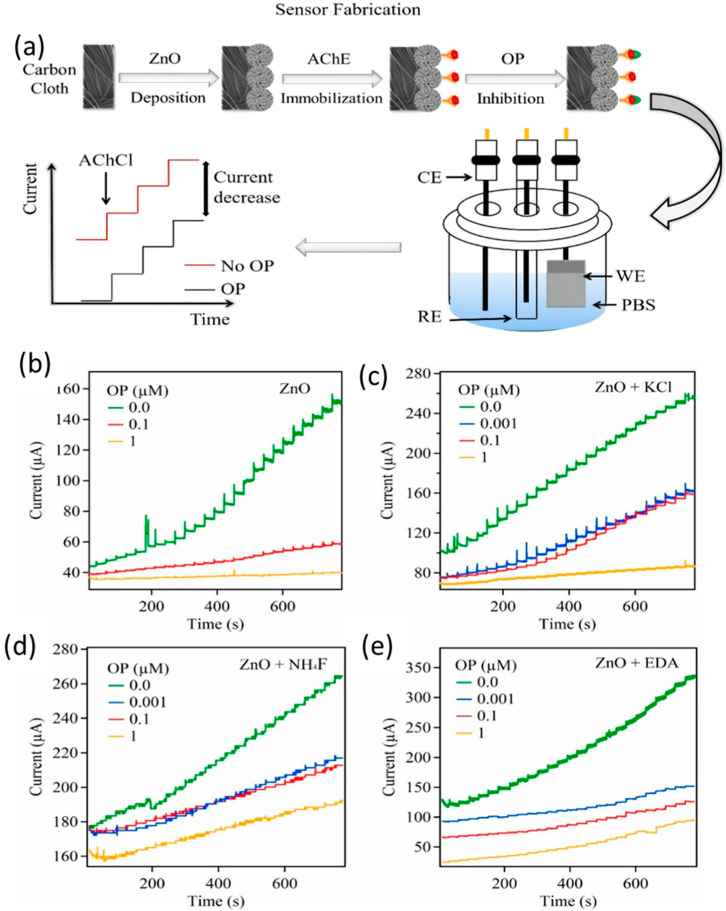
(**a**) Fabrication of metal oxide NPs and the amperometric reaction of ZnO tests deposited on carbon paper. The responses for four unique ZnO tests are compared at (**b**) perfect ZnO, (**c**) ZnO doped with KCl, (**d**) ZnO doped with NH_4_F, and (**e**) ZnO altered with EDA [73].

**Figure 6 nanomaterials-14-01890-f006:**
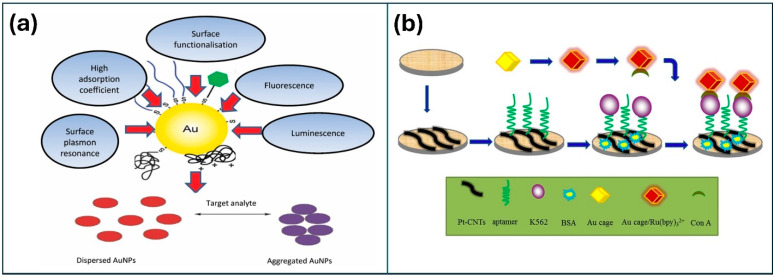
(**a**) Unique optical and chemical properties of AuNPs for biosensors. (**b**) The fabrication process of the ECL to the sensor. Reproduced with permission [76] Copyright 2015 from Elsevier Publications.

**Table 1 nanomaterials-14-01890-t001:** Parameters of CBN materials for detection of various analytes.

Material	Method	Analyte	LOD	Linear Range	Ref
MSN/GO	DPV	CA 153	2.8 × 10^−4^ Um/L	10^−3^–200 U/mL	[49]
MWCNT/	CV	Hepatitis C and tuberculosis genomic DNA	7 fM	0.1 fM–1 pM	[55]
ferrocene					
CNT	DPV	Dopamine (DA)	0.1 μM	0.5–10 μM	[56]
MWCNTs	EIS	Mb	0.08 ng m/L	0.1–90 ng m/L	[57]
SWCNT	DPV	Vibroparahaemolytics thermolabile hemolysin (tlh) gene	7.27 μM	1.0 × 10^−6^–1 × 10^−13^ mol/L	[59]
CNTs	CV	Rutin	0.075 μM	0.10–51 μM	[59]
CNTs	CV	Rutin	0.081 μM	0.10–31 μM	[60]
Graphene	CV, LSV	Puerarin	0.04 μM	0.06–6.0 μM	[61]
Graphene	CV	Puerarin	0.006 μM	0.02–40 μM	[62]
CNTs–graphene	CV	Hyperin	0.001 μM	0.005–1.5 μM	[44]
Graphene quantum dots (GQDs)	CV	Rutin	0.011 μM	0.05–10 μM	[63]
GQD	DPV	Quercentin (Que)	8.2 × 10^−4^ μM	0.002–1.6 μM	[64]
Mesoporous carbon	CV	Rutin	0.002 μM	0.1–30 μM	[65]

Note: CV-cyclic voltammetry, DPV-differential pulse voltammetry, EIS-electrochemical impedance spectroscopy, LSV-linear sweep voltammetry.

**Table 2 nanomaterials-14-01890-t002:** Parameters of MNPs for detection of various analytes.

Material	Method	Analyte	LOD	Linear Range	Ref
Au	Amperometry	Cysteine	3.1 μM	10–80 μM	[81]
		Glutathione	0.1 μM	0.3–10 μM	
		Methionine	1 μM	3.3–39 μM	
		Homocysteine	0.6 μM	2.2–30 μM	
Au	EIS	Carcinoma antigen 125	6.7 U/mL	0–100 U/mL	[82]
		Carcinoma antigen 125	419 ng/mL	450 ng/mL–2.916 μg/mL	
Pt	EIS	Carcinoma antigen 125	386 ng/mL	450 ng/mL–2.916 μg/mL	[83]
Pt	Amperometry	Glucose	44.3 μM	0.25–6.0 mM	[84]
CuO nanospheres	Amperometry	Glucose	1 μM	2.5–20 mM	[85]
ZnCo_2_O_4_	Amperometry	DA	15.5 μM	5–100 μM	[86]
TiO_2_	Amperometry	Glucose	100 nM	100 nM–5 mM	[87]
Pd–Cu	-	Glucose	20 μM	2–18 mM	[88]
Au	-	Glucose	0.05 μM	0.1–25 mM	[89]
Pt	-	Glucose	7.2 × 10^−8^ M	1.0 × 10^−7^–2.0 × 10^−5^ M	[90]

Note: EIS-electrochemical impedance spectroscopy.

**Table 3 nanomaterials-14-01890-t003:** Key parameters of various CBN materials and M/MONPs in electrochemical biosensor applications.

Material	Selectivity	Detection Limit	Sensitivity	Technical Benefits	Application Examples
Graphene & Graphene Oxide (GO)	Selective with modifications (e.g., enzymes, antibodies) for biomolecules like glucose, DNA	Low (fM–pM levels for DNA)	High due to excellent electron transfer properties	High surface area, stability, tunable functional groups	Glucose sensors, DNA biosensors
Carbon Nanotubes (CNTs)	Functionalized for specific biomolecule binding, selective to neurotransmitters, DNA	Low (nM–µM range)	High, due to conductivity and large surface area	Excellent conductivity, stability under diverse conditions	Dopamine, glucose, and DNA sensing
Carbon Dots (CDs)	Modified for ion/molecule targeting, high specificity for small molecules	Low (nM range)	High, with luminescent properties enhancing signal	Stable luminescence, customizable surface functionality	Ion sensing, environmental pollutant detection
Activated Carbon	Porous structure allows selective adsorption, suited for various chemical sensors	Moderate (µM range)	Moderate, depending on surface modifications	High porosity, low cost, good stability	VOCs, pollutant detection
Gold Nanoparticles (AuNPs)	Functionalized with thiol groups, antibodies for specific proteins, DNA	Ultra-low (fM–pM levels)	Very high due to surface plasmon resonance (SPR)	Biocompatible, excellent stability, SPR-enhanced signals	Cancer biomarkers, virus detection
Silver Nanoparticles (AgNPs)	Targeted ion detection, antimicrobial properties	Low (pM–nM levels)	High, SPR enhances sensitivity	Antimicrobial, strong SPR, surface modifiable	Glucose, H_2_O_2_, pathogen detection
Platinum (Pt) & Palladium (Pd) Nanoparticles	Catalytic for reactions (e.g., glucose oxidation), selective in nonenzymatic sensors	Very low (nM–pM for glucose)	Very high due to catalytic efficiency	Catalytic properties, stable under harsh conditions	Nonenzymatic glucose biosensors
Metal Oxides (ZnO, TiO_2_, Fe_2_O_3_)	Selective to gases, ions (e.g., ZnO for pH, TiO_2_ for pollutants)	Moderate to low (µM–nM range)	High with strong response to analyte concentration	Tunable semiconducting properties, stable	pH sensors, pollutant, and gas detection

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
