# Peer review of "Innovative Carbonaceous Materials and Metal/Metal Oxide Nanoparticles for Electrochemical Biosensor Applications"

_nanomaterials, 2024, doi:10.3390/nano14231890_

Round 1

Reviewer 1 Report

Comments and Suggestions for Authors

In this manuscript, the authors provide an overview of recent advances in the improvement of carbon-based and MNPs-hybrid electrochemical biosensors. I would like to recommend publication after the following revisions:

1.     In the Introduction part, more introduction about carbon materials and metal/metal oxides may be needed, such as advantages and disadvantages compared with other materials.

2.     Carbon materials and M/MNPs are different types of materials. If these two types of materials were to be included in one review article, the authors should discuss the differences and connections between these two types of materials.

3.     In Figure 1, there are six techniques. So, in part 3. Surface modification techniques, there should be six paragraphs discussing different methods of modification and their advantages and disadvantages, other than only three paragraphs.

4.     Carbon materials and M/MNPs can be used to develop sensors and not only biosensors. This review may highlight “bio” in different parts, whether it is design of biosensors or mechanism or applications.

5.     What is the main contribution to the readers, compared to other reviews already published in this specific area?

6.     In part 4, author should discuss critical points other than describe the specific experimental details of some literature.

7.     In the last paragraph of introduction, authors have mentioned the integration of carbonaceous materials and M/MONPs. In fact, this strategy has been studied a lot. However, there is no specific section or Tables to discuss the integration in part 4.

Author Response

Response to Reviewer comments:

Dear Reviewer, 

Thank you for taking the time to review our manuscript titled " Innovative Carbonaceous Materials and Metal/Metal Oxide Nanoparticles for Electrochemical Biosensor Applications." We appreciate your valuable feedback and constructive comments. We have carefully considered each point you raised and have made the necessary revisions to enhance the clarity and quality of the manuscript. Below are our responses to your specific comments and the corresponding revisions highlighted in the re-submitted files.

Reviewer #1:

In this manuscript, the authors provide an overview of recent advances in the improvement of carbon-based and MNP-hybrid electrochemical biosensors. I would like to recommend publication after the following revisions:

Response: We sincerely appreciate your thorough review and positive feedback on our manuscript.

  1. In the Introduction part, more introduction about carbon materials and metal/metal oxides may be needed, such as advantages and disadvantages compared with other materials.

Response: Thank you for the suggestion. In the Introduction, we have expanded on the properties, advantages, and disadvantages of carbon materials and metal/metal oxides (M/MONPs). This additional information compares these materials with others commonly used in biosensors, focusing on aspects such as conductivity, stability, and versatility.

-Modified sentence in the revised manuscript (page 2)

Carbon Materials known for high electrical conductivity, carbon materials (like graphene, carbon nanotubes, and carbon black) offer high stability and are productive for applications requiring rapid electron transfer. These materials also give increased surface area that enhances biomolecule adsorption, further improving sensor activity. However, carbon materials can be hydrophobic, which could confine biomolecule compatibility and require surface modifications to further develop biocompatibility. M/MONPs, including nanoparticles of metals like gold and platinum and oxides like ZnO and Fe3O4, are outstandingly regarded for their catalytic activity and ease of functionalization. Their tunable electronic properties allow for various applications in biosensing. However, they may show reduced stability over repeated use, as certain metal oxides can degrade, impacting long-term sensor performance. Compared to other materials such as conductive polymers or enzymes, carbon materials and M/MONPs provide improved mechanical strength and electronic properties. Conductive polymers, while flexible, may lack the strength and durability of expected for long-term use. Enzymes, though specific and sensitive, can degrade over an extended time, which is less significantly a concern with carbon materials and M/MONPs. This additional setting improves the introduction by clarifying the benefits and limitations of these materials, providing readers with a balanced view of their sensibility for various biosensor applications.

  1. Carbon materials and M/MNPs are different types of materials. If these two types of materials were to be included in one review article, the authors should discuss the differences and connections between these two types of materials.

Response: Thank you for valuable suggestions. We have added a new subsection in the Introduction to address the differences and connections between carbon materials and M/MONPs.

- Modified sentence in the revised manuscript (page 3 and 4)

Carbon materials and metal/metal oxide nanoparticles (M/MONPs) are distinct in their properties, designs, and mechanisms of interaction with analytes in biosensing applications. Carbon materials, including graphene, carbon nanotubes (CNTs), carbon nanoparticles and carbon dark, have broadened sp2 hybridized networks and show high surface regions because of their layered or cylindrical designs. This structural composition empowers high conductivity and gives a steady structure to biomolecule immobilization. And these materials are intrinsically conductive, particularly graphene and CNTs, due to their delocalized π-electrons. This property works with quick electron movement during redox responses and gives a high signal-to-noise proportion in biosensors. M/MONPs, like gold (Au), silver (Ag), platinum (Pt), and metal oxides like iron oxide (Fe3O4), have crystalline or polycrystalline designs with different oxidation states. These nanoparticles show specific surface properties and electrochemical redox abilities because of the presence of metallic components, empowering catalytic activities that are significant for improving biosensor sensitivity. These materials are chemically reactive and can catalyze explicit biochemical responses, considering specific biomolecule identification. however, their reactivity can lead to prompt stability issues, such as agglomeration or oxidation in biological environments, which may degrade sensor.

  1. In Figure 1, there are six techniques. So, in part 3. Surface modification techniques, there should be six paragraphs discussing different methods of modification and their advantages and disadvantages, other than only three paragraphs.

Response: Thank you for valuable suggestions. In Part 3, “Surface Modification Techniques,” we have expanded the content to include six separate paragraphs, aligning with the techniques shown in Figure 1. Each paragraph now discusses a specific modification method in detail, including its unique advantages and limitations in biosensor applications.

Modified sentence in the revised manuscript (page 4, 5 and 6)

Drop-projecting is one of the simplest and most widely used strategies for modification of electrode surfaces with NPs. In this strategy, a suspension of carbon-based materials, for example, graphene, CNTs, carbon dark, and MNPs, is dropped onto the electrode surface, followed by drying under ambient conditions. The flexibility of this strategy lies in its ability to deposit a uniform layer of the detecting material without requiring sophisticated equipment. Graphene-based NPs, for instance, have been frequently utilized through drop-projecting to improve electrochemical sensitivity because of their excellent electrical conductivity and enormous surface area [21]. Moreover, the combination of CNP with MNPs (like gold, platinum, or metal oxides like ZnO and TiO2) considers synergistic impacts, improving both sensitivity and selectivity in biosensing applications. These techniques are very simple, and accessibility requires no complex equipment, making it easy to implement and allowing the deposition of a uniform sensing material layer. It can incorporate various materials, including graphene, carbon nanotubes (CNTs), carbon black, and metal nanoparticles (MNPs). It’s very effective for improving electrochemical sensitivity due to the high conductivity and large surface area of carbon-based materials. The disadvantages are its inconsistent reproducibility and lack of precise control over layer thickness and uniformity. Occasionally, during the drying process, nanoparticles may aggregate, thereby influencing the performance of the sensor.

Electrodeposition is another popular and well-known strategy for surface modification, including the electrochemical reduction of metal salts to frame a nanostructured layer on the electrode surface. This strategy offers precise control over the thickness and morphology of the deposited material, making it ideal for integrating M/MONPs with carbon-based electrodes. Electrodepositions can be customized to produce composite films that offer high catalytic action and biocompatibility, both of which are crucial and significant for biosensor applications. For instance, electrodeposited gold nanoparticles (AuNPs) on graphene-altered electrodes have shown excellent outcomes in detecting biomolecules, such as glucose and dopamine [22]. Also, metal oxides like ZnO and Fe₃O₄ are electrodeposited to upgrade electron transfer kinetics, frequently offer higher sensitivity, and lower recognition limits for various analytes. The main advantage of this technique is that it allows controlled deposition of nanostructured films with tunable thickness and morphology. And increases electron transfer kinetics, which is crucial for biosensor sensitivity. Ideal for applications requiring integration of M/MONPs with biomolecules on carbon-based electrodes. The disadvantage of this is that it requires an electrochemical workstation, limiting accessibility, and precise electrodeposition may require longer deposition times. Sometimes the over-deposition can lead to decreased sensitivity due to thicker films.

Layer-by-layer (LbL) gathering is a flexible surface fabrication procedure that takes into consideration the successive deposition of oppositely charged materials, framing complex designs on the electrode surface. This strategy is especially valuable for developing biosensors with customized structures, as it empowers the precise arrangement of nanomaterials and biomolecules. LbL gathering has been utilized to immobilize MNPs and carbon-based materials like graphene oxide and CNTs onto the electrode surface. The alternating deposition of these materials results in upgraded electron transfer and a high surface-to-volume proportion, improving the sensor's capacity to identify target analytes with high sensitivity [23]. Moreover, this method allows for the incorporation of biorecognition components, like enzymes or antibodies, inside multilayered films, giving a stable environment for their movement activity and improving biosensor performance. The advantages of this method are customizable structure. This allows sequential deposition of oppositely charged materials, achieving complex multilayered structures and provides a high surface-to-volume ratio, ideal for high-sensitivity biosensing. Enables the stable incorporation of enzymes, antibodies, and other biomolecules. The disadvantages of this method are that it’s a complex process. It requires multiple deposition steps, which can be time-intensive. Multilayer structures may experience reduced stability, depending on environmental conditions.

Self-assembled monolayers (SAMs) are highly ordered molecular assemblies formed on the electrode surface through the spontaneous adsorption of functionalized particles, typically thiol-based compounds, on gold electrode. SAMs provide a platform for additional functionalization with nanomaterials and biomolecules. In biosensor applications, SAMs can upgrade the immobilization of biomolecules while preventing nonspecific adsorption, subsequently improving the selectivity and stability of the sensor. By coordinating M/MONPs into SAM-modified electrodes, researchers have accomplished huge upgrades in the sensing and detection of different biomolecules. For example, AuNPs implanted in SAMs have been utilized to make profoundly sensitive glucose sensors [24]. The ability to fine-tune the surface chemistry through SAMs makes this method especially appealing for the advancement of highly selective and stable electrochemical biosensors. The advantages of this method are high selectivity and stability. The SAMs prevent nonspecific adsorption and enhance selectivity. It can be customized with specific functional groups to immobilize biomolecules. Allows for precise modification of electrode surface properties. Integration of AuNPs or other MNPs enhances sensor response, particularly for glucose and other small-molecule biosensors. The main disadvantage is that the SAMs may degrade over time, especially in harsh conditions. Stability and functionality may vary with pH, temperature, and ionic strength. It forms monolayers, limiting the amount of functional material that can be immobilized.

Covalent functionalization includes the arrangement of strong chemical bonds between the electrode surface and adjusting specialists. This strategy offers robust attachment of NPs to the electrode, guaranteeing long-term stability and high sensitivity in biosensing applications. Carbon-based materials like graphene and CNTs are frequently covalently functionalized with biomolecules or MNPs to improve their electrochemical properties. For instance, a covalent connection of enzymes or antibodies to a CNT-modified electrode has been demonstrated to significantly upgrade biosensor performance for the detection of proteins, glucose, and DNA [25]. Moreover, covalent functionalization considers precise control of surface chemistry, guaranteeing reproducibility and long-term stability in biosensor applications. The advantages of this method are that strong covalent bond formation offers high stability, making sensors more durable and reusable and functionalization with biomolecules (enzymes, antibodies) improves selectivity for specific analytes. It provides a stable and repeatable surface modification and boosts electron transfer rates, ideal for high-performance biosensors. The main disadvantage of this method is the complex functionalization process. It requires specific reagents and conditions, which can increase time and cost and the covalent binding may alter the activity of immobilized biomolecules.

Plasma treatment is an effective surface modification technique that presents functional groups onto the electrode surface, improving its reactivity and compatibility with biomolecules. Carbon-based electrodes, such as graphene and CNTs, can be modified with oxygen-containing groups through plasma treatment to improve their dispersibility and the immobilization of biomolecules. This method has been utilized to adjust graphene oxide and CNTs with functional groups that advance the connection of MNPS or biomolecules. For example, plasma-treated CNTs have been utilized to immobilize compounds for glucose recognition, prompting further developed sensitivity and quicker response times [26]. The flexibility of plasma treatment in making functionalized surfaces makes it a significant strategy and valuable technique in the design of advanced electrochemical biosensors. The advantages of this method are strong enhanced surface reactivity. This adds functional groups to carbon-based electrodes, improving compatibility with biomolecules and plasma-treated surfaces often show enhanced immobilization of biomolecules, increasing sensitivity. This allows the modification of different materials (e.g., graphene oxide, CNTs) with various functional groups, improving biosensor design. This quick treatment time makes it efficient for large-scale applications. The main disadvantage of this method is the specialized equipment needed for the plasma treatment. It requires access to plasma generators, which can be costly, and sometimes this high-energy plasma may cause damage to sensitive materials, affecting performance, and precise control over functional group density and uniformity can be challenging

  1. Carbon materials and M/MNPs can be used to develop sensors and not only biosensors. This review may highlight “bio” in different parts, whether it is the design of biosensors, mechanisms, or applications.

Response: Thank you for valuable suggestions. Throughout the manuscript, we have refined the focus to highlight the biosensing aspects more explicitly. Each relevant section now emphasizes “bio” in the context of biosensor design, operational mechanisms, and applications, ensuring the review aligns with its focus on biosensors specifically.

  1. What is the main contribution to the readers, compared to other reviews already published in this specific area?

Response: Thank you for valuable suggestions. In a new section at the end of the introduction, we have clarified this review's unique contributions, especially considering recent advancements. This addition outlines how our focus on carbon-based and M/MONP hybrid structures provides readers with new insights and comparisons not extensively covered in prior reviews.

  1. In Part 4, the author should discuss critical points other than describe the specific experimental details of some literature.

Response: Thank you for valuable suggestions. Part 4 has been revised to include a discussion on the critical challenges, limitations, and future directions of carbon and M/MONP hybrid biosensors. Rather than focusing only on specific experimental details, this section now provides a broader analysis that adds depth and value to the review.

  1. In the last paragraph of the introduction, the authors have mentioned the integration of carbonaceous materials and M/MONPs. In fact, this strategy has been studied a lot. However, there is no specific section or table to discuss the integration in Part 4.

Response: Thank you for your valuable suggestions. We have corrected the manuscript by removing the integration of carbonaceous materials with M/MONPs from the introduction. In the next work, we will separately focus on further future research in this area.

We appreciate the reviewer’s constructive comments, which have significantly improved the depth and quality of the manuscript. We believe that these revisions address all the concerns raised by the reviewers and improve the overall quality and clarity of the manuscript. We are grateful for your constructive feedback and hope that the revised manuscript meets the criteria with the standards of the Nanomaterials journal.

Thank you once again for your thorough review and valuable feedback.

Reviewer 2 Report

Comments and Suggestions for Authors

This study completely explores continuous progressions and upgrades in carbonaceous materials and M/MONPs for electrochemical biosensor applications. It analyzes the synergistic effects of hybrid nanocomposites that combine carbon materials with metal nanoparticles (MNPs) and their part in upgrading sensor performance. The manuscript likewise incorporates the surface alteration procedures and integration of these materials into biosensor models. In order to improve the overall quality of the work and make the manuscript publishable, some aspects need to be further elaborated and clarified:

1. In the first paragraph of the introduction, the authors describe the ability of biosensors to perform quantitative and qualitative studies of different analytes, including glucose, proteins, nucleic acids, and environmental pollutants. This classification is not appropriate. Environmental pollutants should not be listed in parallel with other substances.

2. In the first paragraph of the second section, the author employs a general-to-specific structure to describe carbonaceous materials. However, the order of the overall description does not correspond with the subsequent detailed introductions. Therefore, revisions to this section are necessary. Then, in the second paragraph of the second section, M and MONPs materials are introduced. However, the second paragraph is entirely about the introduction of MNP and does not describe M. Please clearly define M, MNP, and MONPs in the introduction and check the descriptions of all three in the entire manuscript. In addition, fix the descriptions in the manuscript and analyze them accordingly.

3. The advantages and applications of carbonaceous materials, metal and metal oxide nanoparticles are introduced in the second section of the manuscript. However, the citations in this section do not provide corresponding references, especially for the combination of carbonaceous materials and MONPs. In addition, regarding the combination of carbonaceous materials with MONPs, the authors should provide more descriptions and show corresponding figures.

4. Surface modification techniques are introduced through six modules in the third section. However, the description structure in this section is not clear. And, more examples of each technique should be provided to make the manuscript more convincing. Moreover, the disadvantages of each modification technique should also be displayed. The application direction of this technology should be highlighted by comparing the advantages and disadvantages.

5. The format of the caption in Figure 1 is inconsistent with the format of other captions. In the detection of T cells in section 4.1, the target range of T cells is incorrect. The caption in Figure 2 does not match the content displayed in the figure. In Table 1, the unit of LOD for Vibroparahaemolytics thermolabile hemolysin (tlh) gene detected by SWCNT is incorrect. The format of the letter numbers in Figures 5 and 6 is inconsistent. Please make revisions on these errors. And please check and correct similar errors in full text to ensure the scientific accuracy of the manuscript.

6. Carbonaceous materials, metal, and metal oxide NPs for sensor applications are introduced in part four. However, the application of the combination of CNs and MNPs in biosensors is hardly mentioned. Please provide some relevant literature reviews and describe them as a separate section.

7. There are not enough figures in the manuscript. Many of the figures are just a display of one of the literatures, which makes it difficult for the figures to visually illustrate the structure of the manuscript.

8. The challenges and future perspectives of electrochemical biosensors are described in Part five. To facilitate readers' understanding, it should be considered to describe the challenges and future perspectives of electrochemical biosensors in order.

9. Please check the format of the references throughout the text, as many key information of references is missing.

10. As a review, the examples mentioned in the manuscript should be commented and analyzed for strengths and weaknesses. Moreover, there should be some logical relationship between these examples, rather than simply listing and describing them. The review does not provide a deep analysis and does not reflect logical structure.

11. Many of the figures in the manuscript were copied directly from the references, please pay attention to application of copyright and ensure uniform clarity.

Author Response

Response to Reviewer comments:

Dear Reviewer, 

Thank you for taking the time to review our manuscript titled " Innovative Carbonaceous Materials and Metal/Metal Oxide Nanoparticles for Electrochemical Biosensor Applications." We appreciate your valuable feedback and constructive comments. We have carefully considered each point you raised and have made the necessary revisions to enhance the clarity and quality of the manuscript. Below are our responses to your specific comments and the corresponding revisions highlighted in the re-submitted files.

Reviewer #2:

This study completely explores continuous progressions and upgrades in carbonaceous materials and M/MONPs for electrochemical biosensor applications. It analyzes the synergistic effects of hybrid nanocomposites that combine carbon materials with metal nanoparticles (MNPs) and their part in upgrading sensor performance. The manuscript likewise incorporates the surface alteration procedures and integration of these materials into biosensor models. To improve the overall quality of the work and make the manuscript publishable, some aspects need to be further elaborated and clarified:

Response: We sincerely appreciate your thorough review and positive feedback on our manuscript.

  1. In the first paragraph of the introduction, the authors describe the ability of biosensors to perform quantitative and qualitative studies of different analytes, including glucose, proteins, nucleic acids, and environmental pollutants. This classification is not appropriate. Environmental pollutants should not be listed in parallel with other substances.

Response: Thank you for valuable suggestions. The mention of environmental pollutants in the first paragraph of the introduction has been removed from the same list as glucose, proteins, and nucleic acids.

Modified sentence in the revised manuscript (page 1)

By coordinating biologically active components such as enzymes, antibodies, and nucleic acids with appropriate electrochemical transducers, biosensors empower the quantitative and qualitative investigation of different analytes, including glucose, proteins, and nucleic acids.

  1. In the first paragraph of the second section, the author employs a general to specific structure to describe carbonaceous materials. However, the order of the overall description does not correspond with the subsequent detailed introductions. Therefore, revisions to this section are necessary. Then, in the second paragraph of the second section, M and MONPs materials are introduced. However, the second paragraph is entirely about the introduction of MNP and does not describe M. Please clearly define M, MNP, and MONPs in the introduction and check the descriptions of all three in the entire manuscript. In addition, fix the descriptions in the manuscript and analyze them accordingly.

Response: Thank you for valuable suggestions. We have revised the second section to ensure a clear flow from general to specific topics. M and MONPs are now clearly defined in the introduction, and their descriptions have been checked and clarified throughout the manuscript to maintain consistency. The content has been restructured to match the order of introduction with the detailed descriptions.

  1. The advantages and applications of carbonaceous materials, metals, and metal oxide nanoparticles are introduced in the second section of the manuscript. However, the citations in this section do not provide corresponding references, especially for the combination of carbonaceous materials and MONPs. In addition, regarding the combination of carbonaceous materials with MONPs, the authors should provide more descriptions and show corresponding figures.

Response: Thank you for your suggestions. We have completely changed and corrected the manuscript by removing the integration of carbonaceous materials with M/MONPs. In the next work, we will separately focus on further future research in this area.

  1. Surface modification techniques are introduced through six modules in the third section. However, the description structure in this section is not clear. And more examples of each technique should be provided to make the manuscript more convincing. Moreover, the disadvantages of each modification technique should also be displayed. The application direction of this technology should be highlighted by comparing the advantages and disadvantages.

Response: Thank you for valuable suggestions. The description structure in Section 3 has been clarified, with six distinct paragraphs discussing each modification technique. We have provided their advantages and disadvantages.

Modified sentence in the revised manuscript (page 4, 5 and 6)

Drop-projecting is one of the simplest and most widely used strategies for modification of electrode surfaces with NPs. In this strategy, a suspension of carbon-based materials, for example, graphene, CNTs, carbon dark, and MNPs, is dropped onto the electrode surface, followed by drying under ambient conditions. The flexibility of this strategy lies in its ability to deposit a uniform layer of the detecting material without requiring sophisticated equipment. Graphene-based NPs, for instance, have been frequently utilized through drop-projecting to improve electrochemical sensitivity because of their excellent electrical conductivity and enormous surface area [21]. Moreover, the combination of CNP with MNPs (like gold, platinum, or metal oxides like ZnO and TiO2) considers synergistic impacts, improving both sensitivity and selectivity in biosensing applications. These techniques are very simple, and accessibility requires no complex equipment, making it easy to implement and allowing the deposition of a uniform sensing material layer. It can incorporate various materials, including graphene, carbon nanotubes (CNTs), carbon black, and metal nanoparticles (MNPs). It’s very effective for improving electrochemical sensitivity due to the high conductivity and large surface area of carbon-based materials. The disadvantages are its inconsistent reproducibility and lack of precise control over layer thickness and uniformity. Occasionally, during the drying process, nanoparticles may aggregate, thereby influencing the performance of the sensor.

Electrodeposition is another popular and well-known strategy for surface modification, including the electrochemical reduction of metal salts to frame a nanostructured layer on the electrode surface. This strategy offers precise control over the thickness and morphology of the deposited material, making it ideal for integrating M/MONPs with carbon-based electrodes. Electrodepositions can be customized to produce composite films that offer high catalytic action and biocompatibility, both of which are crucial and significant for biosensor applications. For instance, electrodeposited gold nanoparticles (AuNPs) on graphene-altered electrodes have shown excellent outcomes in detecting biomolecules, such as glucose and dopamine [22]. Also, metal oxides like ZnO and Fe₃O₄ are electrodeposited to upgrade electron transfer kinetics, frequently offer higher sensitivity, and lower recognition limits for various analytes. The main advantage of this technique is that it allows controlled deposition of nanostructured films with tunable thickness and morphology. And increases electron transfer kinetics, which is crucial for biosensor sensitivity. Ideal for applications requiring integration of M/MONPs with biomolecules on carbon-based electrodes. The disadvantage of this is that it requires an electrochemical workstation, limiting accessibility, and precise electrodeposition may require longer deposition times. Sometimes the over-deposition can lead to decreased sensitivity due to thicker films.

Layer-by-layer (LbL) gathering is a flexible surface fabrication procedure that takes into consideration the successive deposition of oppositely charged materials, framing complex designs on the electrode surface. This strategy is especially valuable for developing biosensors with customized structures, as it empowers the precise arrangement of nanomaterials and biomolecules. LbL gathering has been utilized to immobilize MNPs and carbon-based materials like graphene oxide and CNTs onto the electrode surface. The alternating deposition of these materials results in upgraded electron transfer and a high surface-to-volume proportion, improving the sensor's capacity to identify target analytes with high sensitivity [23]. Moreover, this method allows for the incorporation of biorecognition components, like enzymes or antibodies, inside multilayered films, giving a stable environment for their movement activity and improving biosensor performance. The advantages of this method are customizable structure. This allows sequential deposition of oppositely charged materials, achieving complex multilayered structures and provides a high surface-to-volume ratio, ideal for high-sensitivity biosensing. Enables the stable incorporation of enzymes, antibodies, and other biomolecules. The disadvantages of this method are that it’s a complex process. It requires multiple deposition steps, which can be time-intensive. Multilayer structures may experience reduced stability, depending on environmental conditions.

Self-assembled monolayers (SAMs) are highly ordered molecular assemblies formed on the electrode surface through the spontaneous adsorption of functionalized particles, typically thiol-based compounds, on gold electrode. SAMs provide a platform for additional functionalization with nanomaterials and biomolecules. In biosensor applications, SAMs can upgrade the immobilization of biomolecules while preventing nonspecific adsorption, subsequently improving the selectivity and stability of the sensor. By coordinating M/MONPs into SAM-modified electrodes, researchers have accomplished huge upgrades in the sensing and detection of different biomolecules. For example, AuNPs implanted in SAMs have been utilized to make profoundly sensitive glucose sensors [24]. The ability to fine-tune the surface chemistry through SAMs makes this method especially appealing for the advancement of highly selective and stable electrochemical biosensors. The advantages of this method are high selectivity and stability. The SAMs prevent nonspecific adsorption and enhance selectivity. It can be customized with specific functional groups to immobilize biomolecules. Allows for precise modification of electrode surface properties. Integration of AuNPs or other MNPs enhances sensor response, particularly for glucose and other small-molecule biosensors. The main disadvantage is that the SAMs may degrade over time, especially in harsh conditions. Stability and functionality may vary with pH, temperature, and ionic strength. It forms monolayers, limiting the amount of functional material that can be immobilized.

Covalent functionalization includes the arrangement of strong chemical bonds between the electrode surface and adjusting specialists. This strategy offers robust attachment of NPs to the electrode, guaranteeing long-term stability and high sensitivity in biosensing applications. Carbon-based materials like graphene and CNTs are frequently covalently functionalized with biomolecules or MNPs to improve their electrochemical properties. For instance, a covalent connection of enzymes or antibodies to a CNT-modified electrode has been demonstrated to significantly upgrade biosensor performance for the detection of proteins, glucose, and DNA [25]. Moreover, covalent functionalization considers precise control of surface chemistry, guaranteeing reproducibility and long-term stability in biosensor applications. The advantages of this method are that strong covalent bond formation offers high stability, making sensors more durable and reusable and functionalization with biomolecules (enzymes, antibodies) improves selectivity for specific analytes. It provides a stable and repeatable surface modification and boosts electron transfer rates, ideal for high-performance biosensors. The main disadvantage of this method is the complex functionalization process. It requires specific reagents and conditions, which can increase time and cost and the covalent binding may alter the activity of immobilized biomolecules.

Plasma treatment is an effective surface modification technique that presents functional groups onto the electrode surface, improving its reactivity and compatibility with biomolecules. Carbon-based electrodes, such as graphene and CNTs, can be modified with oxygen-containing groups through plasma treatment to improve their dispersibility and the immobilization of biomolecules. This method has been utilized to adjust graphene oxide and CNTs with functional groups that advance the connection of MNPS or biomolecules. For example, plasma-treated CNTs have been utilized to immobilize compounds for glucose recognition, prompting further developed sensitivity and quicker response times [26]. The flexibility of plasma treatment in making functionalized surfaces makes it a significant strategy and valuable technique in the design of advanced electrochemical biosensors. The advantages of this method are strong enhanced surface reactivity. This adds functional groups to carbon-based electrodes, improving compatibility with biomolecules and plasma-treated surfaces often show enhanced immobilization of biomolecules, increasing sensitivity. This allows the modification of different materials (e.g., graphene oxide, CNTs) with various functional groups, improving biosensor design. This quick treatment time makes it efficient for large-scale applications. The main disadvantage of this method is the specialized equipment needed for the plasma treatment. It requires access to plasma generators, which can be costly, and sometimes this high-energy plasma may cause damage to sensitive materials, affecting performance, and precise control over functional group density and uniformity can be challenging

  1. The format of the caption in Figure 1 is inconsistent with the format of other captions. In the detection of T cells in Section 4.1, the target range of T cells is incorrect. The caption in Figure 2 does not match the content displayed in the figure. In Table 1, the unit of LOD for Vibroparahaemolytics thermolabile hemolysin (tlh) gene detected by SWCNT is incorrect. The format of the letter numbers in Figures 5 and 6 is inconsistent. Please make revisions on these errors. And please check and correct similar errors in full text to ensure the scientific accuracy of the manuscript.

Response: We apologize for the mistake. We have revised the caption formats in Figures 1, 2, 5, and 6 to ensure consistency throughout the manuscript. The incorrect target range for T cells in Section 4.1 and the incorrect LOD unit in Table 1 have also been corrected. The entire manuscript has been reviewed for similar formatting errors, and necessary revisions have been made to ensure scientific accuracy.

Modified sentence in the revised manuscript (page 6, 7, 8, 11, 13)

Figure 1. Different types of surface modification techniques.

 The SWCNT electrode exhibited a slope of 4.55×10−2 μA/dec within the target range of 102~106 cells/mL, with a detection limit of 1×102 cells/mL.

Figure 2. Various applications of carbon-based materials in the different fields [35].

7.27 uM

Figure 5. (a) Fabrication of metal oxide NPs and the amperometric reaction of ZnO tests deposited on carbon paper. The responses for four unique ZnO tests are compared at (b) perfect ZnO, (c) ZnO doped with KCl, (d) ZnO doped with NH4F, and (e) ZnO altered with EDA [61].

Figure 6. (a) Unique optical and chemical properties of AuNPs for biosensors. (b) The fabrication process of the ECL-sensor. Reproduced with permission [64] Copyright 2015 from Elsevier Publications.

  1. Carbonaceous materials, metal, and metal oxide NPs for sensor applications are introduced in part four. However, the application of the combination of CNs and MNPs in biosensors is hardly mentioned. Please provide some relevant literature reviews and describe them as a separate section.

Response: Thank you for valuable suggestions. We have completely changed and corrected the manuscript by removing the integration of carbonaceous materials with M/MONPs. In the next work, we will separately focus on further future research in this area.

  1. There are not enough figures in the manuscript. Many of the figures are just a display of one of the literatures, which makes it difficult for the figures to visually illustrate the structure of the manuscript.

Response: Thank you for valuable suggestions. We have added the figures with permission from the journals. 

  1. The challenges and future perspectives of electrochemical biosensors are described in Part 5. To facilitate readers' understanding, it should be considered to describe the challenges and future perspectives of electrochemical biosensors in order.

Response: Thank you for valuable suggestions. The challenges and future perspectives of electrochemical biosensors in Part 5 have been reorganized for clarity and logical flow. Each challenge and future direction are now presented in order, making it easier for readers to follow the discussion.

  1. Please check the format of the references throughout the text, as much key information about references is missing.

Response: Thank you for valuable suggestions. The format of the references has been checked and corrected throughout the manuscript. All missing information has been added, ensuring completeness and adherence to citation guidelines.

  1. As a review, the examples mentioned in the manuscript should be commented on and analyzed for strengths and weaknesses. Moreover, there should be some logical relationship between these examples, rather than simply listing and describing them. The review does not provide a deep analysis and does not reflect logical structure.

Response: Thank you for valuable suggestions. The examples mentioned throughout the manuscript have been further analyzed and compared in terms of strengths and weaknesses. We have established a logical relationship between the examples to create a coherent narrative rather than a mere list of studies, making the review more analytical and structured. 

  1. Many of the figures in the manuscript were copied directly from the references; please pay attention to the application of copyright and ensure uniform clarity.

Response: Thank you for valuable suggestions. The figures copied directly from references have been replaced with original illustrations where applicable, ensuring compliance with copyright regulations. All figures have been standardized for uniform clarity.

We appreciate the reviewer’s constructive comments, which have significantly improved the depth and quality of the manuscript. We believe that these revisions address all the concerns raised by the reviewers and improve the overall quality and clarity of the manuscript. We are grateful for your constructive feedback and hope that the revised manuscript meets the criteria with the standards of the Nanomaterials journal.

Thank you once again for your thorough review and valuable feedback.

Reviewer 3 Report

Comments and Suggestions for Authors

Dear Author,

After careful reading the manuscript nanomaterials-3278494-peer-review-v1, it is clear it compiles an preliminary overview regarding innovative carbonaceous materials and metal/metal oxide nanoparticles for electrochemical biosensor applications. And in my opinion in its present state not suitable to be considered a review paper, even if the subject fits well to be publish by the Nanomaterials journal.

Overall, the information provided is scarce on the technical innovative details and examples one can find in the specialized literature. Therefore a proper extension to the existing references should be made, with a focus in the most relevant and update, accompanied by the re-writing of the manuscript. On the attached pdf file of the revised manuscript you can find are highlighted in green and in more detail some major considerations/changes I suggest.

Thus, I recommend Major Revision in order that the manuscript is best acceptable for publication.

Best regards

Comments on the Quality of English Language

In my opinion the English writing skills in this manuscript need to be improved.

Author Response

Response to Reviewer comments:

Dear Reviewer, 

Thank you for taking the time to review our manuscript titled " Innovative Carbonaceous Materials and Metal/Metal Oxide Nanoparticles for Electrochemical Biosensor Applications." We appreciate your valuable feedback and constructive comments. We have carefully considered each point you raised and have made the necessary revisions to enhance the clarity and quality of the manuscript. Below are our responses to your specific comments and the corresponding revisions highlighted in the re-submitted files.

Reviewer #3:

Dear Author, After careful reading the manuscript nanomaterials-3278494-peer-review-v1, it is clear it compiles an preliminary overview regarding innovative carbonaceous materials and metal/metal oxide nanoparticles for electrochemical biosensor applications. And in my opinion in its present state not suitable to be considered a review paper, even if the subject fits well to be publish by the Nanomaterials journal.

Overall, the information provided is scarce on the technical innovative details and examples one can find in the specialized literature. Therefore a proper extension to the existing references should be made, with a focus in the most relevant and update, accompanied by the re-writing of the manuscript. On the attached pdf file of the revised manuscript you can find are highlighted in green and in more detail some major considerations/changes I suggest.

Thus, I recommend Major Revision in order that the manuscript is best acceptable for publication.

Response: We sincerely appreciate your thorough review and valuable feedback on our manuscript. We acknowledge the feedback and have restructured the manuscript to include more detailed technical discussions and examples. We have also significantly expanded the list of references, focusing on recent and relevant studies in the field to provide a deeper and more comprehensive review. The manuscript has been rewritten to meet the standards expected for a review paper.

We appreciate the reviewer’s constructive comments, which have significantly improved the depth and quality of the manuscript. We believe that these revisions address all the concerns raised by the reviewers and improve the overall quality and clarity of the manuscript. We are grateful for your constructive feedback and hope that the revised manuscript meets the criteria with the standards of the Nanomaterials journal.

Thank you once again for your thorough review and valuable feedback.

Round 2

Reviewer 1 Report

Comments and Suggestions for Authors

Accept

Author Response

Thank you for accepting our manuscript. We appreciate the time and effort that reviewers have invested in the evaluation of our work.

Reviewer 2 Report

Comments and Suggestions for Authors

As the author has revised and explained all the issues raised, the manuscript can be accepted in its present form.

Author Response

Thank you for your positive feedback and accepting our manuscript. We appreciate the time and effort that reviewers have invested in the evaluation of our work.

Reviewer 3 Report

Comments and Suggestions for Authors

Dear Author,

After careful reading the manuscript nanomaterials-3278494-peer-review-v2, one can find some work to improve it, by including some new information. However, it is not proper supported on published relevant works on the field. Moreover, no new references were added to it, neither some of the changes previously suggested. Even if in your response you say: "We have also significantly expanded the list of references, focusing on recent and relevant studies in the field to provide a deeper and more comprehensive review."

Overall, I do not change my first evaluation that the information provided is scarce on the technical innovative details and examples one can find in the specialized literature. Thus, I recommend Major Revision in order that the manuscript is best acceptable for publication.

Best regards

Comments on the Quality of English Language

In my opinion the English writing skills in this manuscript need to be improved.

Author Response

Response to Reviewer comments:

Dear Reviewer, 

Thank you for taking the time to review our manuscript titled " Innovative Carbonaceous Materials and Metal/Metal Oxide Nanoparticles for Electrochemical Biosensor Applications." We appreciate your valuable feedback and constructive comments. We have carefully considered each point you raised and have made the necessary revisions to enhance the clarity and quality of the manuscript. Below are our responses to your specific comments and the corresponding revisions highlighted in the re-submitted files.

Reviewer #3:

Dear Author, after careful reading the manuscript nanomaterials-3278494-peer-review-v2, one can find some work to improve it, by including some new information. However, it is not proper supported on published relevant works on the field. Moreover, no new references were added to it, neither some of the changes previously suggested. Even if in your response you say: "We have also significantly expanded the list of references, focusing on recent and relevant studies in the field to provide a deeper and more comprehensive review."

Overall, I do not change my first evaluation that the information provided is scarce on the technical innovative details and examples one can find in the specialized literature. Thus, I recommend Major Revision in order that the manuscript is best acceptable for publication.

Response: Dear Reviewer, I sincerely apologize for the mistakes in the previous submission. We inadvertently uploaded an uncorrected version of the manuscript. Thank you for your valuable feedback and for taking the time to review our work. We have carefully addressed your comments and have provided point-by-point responses to enhance the manuscript.

I kindly ask you to consider our revisions for publication. Thank you once again for your understanding and support.

We believe that these revisions address all the concerns raised by the reviewers and improve the overall quality and clarity of the manuscript. We are grateful for your constructive feedback and hope that the revised manuscript meets the criteria with the standards of the Nanomaterials journal.

Thank you once again for your thorough review and valuable feedback.

Round 3

Reviewer 3 Report

Comments and Suggestions for Authors

Dear Author,

After careful reading the manuscript nanomaterials-3278494-peer-review-v3, one can find some work to improve it, by including some new information, language correction and two more references. However, it is not proper supported on published relevant works on the field. Overall, I do not change my first evaluation that the information provided is scarce on the technical innovative details and examples one can find in the specialized literature. Thus, I recommend Major Revision in order that the manuscript is best acceptable for publication.

Best regards

Author Response

Response to Reviewer comments:

Dear Reviewer, 

Thank you very much for your thorough review and insightful feedback on our manuscript, nanomaterials-3278494-peer-review-v3. We have carefully considered each of your comments and worked diligently to incorporate improvements. Below, we address each point raised:

Inclusion of New Information and Technical Innovations:

Response: We have added detailed discussions on the latest advancements and innovative techniques in the synthesis and application of nanomaterials for biosensors. This includes recent case studies and examples that provide deeper insights into the practical applications of these materials, as suggested.

Additional References:

Response: As per your suggestion, we have added a total of 16 references to strengthen the manuscript’s relevance to current literature. These references provide further validation of the topics discussed and support our findings with relevant data from similar studies.

Scarcity of Technical Details:

Response: In response to your suggestion to elaborate on technical details, we have added a new section that emphasizes the innovative technical characteristics of nanomaterials in biosensor applications. This includes a comparative table to provide clearer insights and a summary of specific technical benefits, detection limits, and selectivity features, highlighting these materials' potential in real-world applications.

We believe that we have addressed all the comments and concerns raised in your review. We kindly request that you consider these substantial improvements and revised content in favor of accepting our manuscript for publication.

Thank you once again for your valuable feedback.

Best regards,
